# Ten-Year in-Hospital Mortality Trends among Paediatric Injured Patients in Japan: A Nationwide Observational Study

**DOI:** 10.3390/jcm9103273

**Published:** 2020-10-12

**Authors:** Chiaki Toida, Takashi Muguruma, Masayasu Gakumazawa, Mafumi Shinohara, Takeru Abe, Ichiro Takeuchi, Naoto Morimura

**Affiliations:** 1Department of Disaster Medical Management, The University of Tokyo, 7-3-1 Hongo, Bunkyo-ku, Tokyo 113-8655, Japan; molimula-tky@umin.ac.jp; 2Department of Emergency Medicine, Yokohama City University Graduate School of Medicine, 4-57 Urafunecho, Minami-ku, Yokohama 232-0024, Japan; mgrmtks@gmail.com (T.M.); gakumazawa-tuk@umin.ac.jp (M.G.); shinoharamafumi@yahoo.co.jp (M.S.); abet@yokohama-cu.ac.jp (T.A.); takeqq@yokohama-cu.ac.jp (I.T.)

**Keywords:** paediatric patient, trauma, in-hospital mortality, injury surveillance, injury prevention

## Abstract

Injury is a major cause of worldwide child mortality. This retrospective nationwide study aimed to evaluate the characteristics of paediatric injured patients in Japan and their in-hospital mortality trends from 2009 to 2018. Injured patients aged <17 years were enrolled. Data were extracted from the Japan Trauma Data Bank. In the Cochran-Armitage test, in-hospital mortality significantly decreased during the study period (*p* < 0.001), except among patients <1 year old, and yearly reductions were observed among those with an Injury Severity Score ≥16 and survival rate ≥50% (*p* < 0.001). In regression analyses, patients who underwent urgent blood transfusion within 24 h after hospital admission (odds ratio (OR) = 3.24, 95% confidence interval (CI) = 2.38–4.41) had a higher in-hospital mortality risk. Higher survival probability as per the Trauma and Injury Severity Score was associated with lower in-hospital mortality (OR = 0.92, 95% CI = 0.91–0.92), a risk which decreased from 2009 to 2018 (OR = 6.16, 95% CI = 2.94–12.88). Based on our results, there is a need for improved injury surveillance systems for establishment of injury prevention strategies along with evaluation of the quality of injury care and outcome measures.

## 1. Introduction

Injury, due to road traffic crash, fall, drowning, fires, or poisoning, is a major cause of child mortality worldwide [1]. Previous studies showed that in-hospital mortality of paediatric injured patients varied from 0.19% to 8% in Australia, USA, Canada, and Europe [2,3,4,5,6]. In these reports, the incidence of childhood injury and risk factors for in-hospital mortality after injury also varied by countries. On the one hand, the Centers for Disease Control and Prevention reported that around 12,000 children aged between 0 to 19 years died from injuries each year in the United States, of which the leading cause was road traffic crash [3]. On the other hand, in Japan, the proportion of deaths related to road traffic crash decreased year by year, while the proportion of deaths related to injury events happening at home increased [7,8]. Furthermore, the trend of in-hospital mortality in paediatric injured patients was also reported to vary by injury mechanism and age-related injury characteristics [1,2,3,4,5,6].

Therefore, injury surveillance is essential for the identification of age-related injury characteristics, monitoring of injury trends over time, and estimation of the effectiveness of injury prevention measures [1]. In Japan, the Japan Trauma Data Bank (JTDB) was established in 2003 by the Japanese Association for the Surgery of Trauma and Japanese Association for Acute Medicine with the aim of improving and ascertaining the quality of injury care [9]. Previous studies using JTDB datasets have shown that the in-hospital mortality of adult injured patients have significantly decreased over the last 20 years [10,11], which might be associated with the provision of the Japan Advanced Trauma Evaluation and care (JATEC) off-the-job trauma training courses and trauma education based on standardised trauma treatment guidelines, as developed by the Advanced Trauma Evaluation and Care Association, initiated in 2002 [12]. However, among paediatric injured patients <15 years, the in-hospital mortality did not change after the introduction of the off-the-job training courses named the JATEC in 2002 and Japan Prehospital Trauma Evaluation and Care (JPTEC) in 2003 [11,12,13].

Because of the lack of data on epidemiology and mortality of paediatric injured patients, the trend of in-hospital mortality and the mortality risk in paediatric injured patients in Japan remains unclear. Therefore, this study aimed to evaluate the characteristics and in-hospital mortality of Japanese paediatric injured patients during the 10-year period between 2009 and 2018 for establishment of effective injury prevention measures.

## 2. Materials and Methods

### 2.1. Study Setting and Population

This retrospective observational study was conducted based on data obtained from the JTDB, which registers data on patients with injury and/or burn, and records prehospitalisation and hospital-related information. JTDB data include demographics, comorbidities, injury types, mechanism of injury, means of transportation, vital signs, Abbreviated Injury Scale (AIS) score, prehospital/in-hospital procedures, injury diagnosis as indicated by the AIS, and clinical outcomes. In most cases, physicians trained in AIS coding undertake the online registration of individual patients’ data. JTDB data collection started from 55 hospitals in 2003. The number of participating hospitals increased year by year, up to a total of 280 hospitals in all 47 prefectures in Japan, including 92% of Japanese government-approved tertiary emergency medical centres in March 2019. The Japan Association for the Surgery of Trauma permits open access and updating of existing medical information, and the Japan Association for Acute Medicine evaluates the submitted data [9].

In this study, we used a JTDB dataset that included information from 1 January 2009 to 31 December 2018, which initially yielded data on 313,643 patients. Inclusion criteria for this study were presence of injury and age <17 years. Patients with burns, age >17 years, or with missing key data were excluded from this study. Figure 1 presents a flow diagram of the patients’ disposition in this study. Appendix A presents the trend in missing data rates during the study period. There were no significant statistical differences in the missing data rates by year using a Cochran-Armitage test.

### 2.2. Data Collection

We collected the following information from the JTDB: demographic data (age (years), sex, transportation method, mechanism of injury), clinical parameters (presence of cardiac arrest on hospital arrival, requirement of urgent blood transfusion within 24 h from hospital arrival, AIS value, AIS of the injured region, Revised Trauma Score made of: age, Glasgow Coma Scale, systolic blood pressure, and respiratory rate [14], Injury Severity Score (ISS) [15], survival probability), and outcomes (in-hospital mortality, standardised mortality ratio (SMR)). Outcome measures were SMR and in-hospital mortality trends and risk factors of in-hospital mortality. The survival probability and predicted mortality were calculated using the Trauma and Injury Severity Score (TRISS) [16], and SMR was calculated by dividing the in-hospital mortality by the mean predicted mortality.

### 2.3. Statistical Analysis

This study estimated: (1) patients’ demographic characteristics and outcomes employed during the 10-year study period, (2) 10-year SMR and in-hospital mortality trends, and (3) risk factors associated with in-hospital mortality over 10 years. In the primary analysis, conducted for the identification of characteristics of paediatric injured patients during the study period, a Mann–Whitney U test and Kruskal–Wallis test were used for the analyses of continuous variables, whereas a chi-square test was used for categorical variables. In the secondary analysis, conducted for the identification of the SMR and in-hospital mortality trends during the study period, the Cochran-Armitage test was used to test for trends in SMR and in-hospital mortality by study year, which was treated as an ordinal variable. The following variables were applied to the multivariate logistic regression analyses: age, probability of survival, requirement of urgent blood transfusion, and year. The dependent variable in multivariate logistic regression was in-hospital mortality. In addition, the Cochran-Armitage test and multivariate logistic regression analyses were repeated for injured patients with an ISS ≥16 and those with a survival probability rate ≥50%. The results of these comparisons are expressed as the medians and value ranges (5th–95th percentile) for continuous variables and as patients’ number and percentages for categorical variables. All statistical analyses were performed using STATA/SE software, version 16.0 (StataCorp; College Station, TX, USA). A two-tailed *p*-value < 0.05 indicated statistical significance.

### 2.4. Ethics Statement

This study was approved by the institutional ethics committees of Yokohama City University Medical Centre (approval no. B170900003). The approving authority for data access was the Japanese Association for the Surgery of Trauma (Trauma Registry Committee). Requirement of informed consent from the patients was waived due to the observational nature of the study design.

## 3. Results

During the 10-year study period, data on 16,068 paediatric injured patients were examined (Figure 1). These patients were categorised into the following age groups: <1 year (*n* = 346, 2%), 1–5 years (*n* = 2301, 14%), 6–12 years (*n* = 6093, 38%), and 13–17 years (*n* = 7328, 46%). The median age of the total cohort was 12 years (5th–95th percentile, 2–17). The study sample included patients with blunt injury (*n* = 15,866, 99%), including those with polytrauma (*n* = 2312, 14%). The number of patients with an ISS ≥16 and a survival probability ≥50% were 5787 and 15,407. The overall in-hospital mortality was 4.1 % (*n* = 662), that of patients with an ISS ≥ 16 was 10.9% (*n* = 628), and that of patients with a survival probability ≥ 50% was 0.9% (*n* = 143). The overall SMR was 0.72.

Table 1 shows patients’ demographic, clinical, and outcome data by year. There was no significant difference in median age during the study period. With regards to the injury mechanisms of blunt injury, the rate of road traffic crashes decreased (from 59% in 2009 to 45% in 2018, *p* < 0.001) while that of fall increased (from 24% in 2008 to 26% in 2018, *p* < 0.001), and the percentage of patients with polytrauma decreased (from 15% in 2009 to 12% in 2018, *p* < 0.001). The median ISS decreased (from 10 in 2009 to 9 in 2018, *p* < 0.001) and the survival probability increased (from 99.1% in 2009 to 99.3% in 2018, *p* < 0.001). There was no significant difference in the percentage of patients with an ISS ≥ 16 and with a survival probability ≥ 50% over 10 years.

The 10-year SMR and in-hospital mortality trends during the 10-year study period are shown in Table 1 and Figure 2. In the Cochran-Armitage test, the SMR and overall in-hospital mortality showed significant decreases over the 10-year period (from 0.88 in 2009 to 0.58 in 2018, *p* = 0.006, and from 6.0% in 2009 to 2.9% in 2018, *p* < 0.001, Table 1 and Figure 2). Similarly, the above parameters showed significant reduction among patients with an ISS ≥16 and a survival probability ≥50% (from 14.3% in 2009 to 8.9% in 2018, *p* < 0.001, and from 2.1% in 2009 to 0.5% in 2018, *p* < 0.001, respectively).

In the Cochran-Armitage test, overall in-hospital mortality showed significant decreases over the 10-year period (from 6.0% in 2009 to 2.9% in 2018, *p* < 0.001). Similarly, the values among those with an ISS ≥16 and a TRISS Ps ≥50% showed significant decreases (from 14.3% in 2009 to 8.9% in 2018, *p* < 0.001, and from 2.1% in 2009 to 0.5% in 2018, *p* < 0.001, respectively).

Figure 3 shows the in-hospital mortality for each age group. Although decreases were observed in the in-hospital mortality of patients aged 1–5 years, 6–12 years, and 13–17 years (from 9.5% in 2009 to 3.9% in 2018, *p* = 0.006, from 4.5% in 2009 to 1.0% in 2018, *p* < 0.001, and from 6.1% in 2009 to 3.6% in 2018, *p* < 0.001, respectively), there were no significant changes in in-hospital mortality of those <1 year old.

All variables expressed the number of death (mortality, %) in the Figure. In the Cochran-Armitage test, reductions were observed in the in-hospital mortality values of patients aged 1–5 years, 6–12 years, and 13–17 years (from 9.5% in 2009 to 3.9% in 2018, *p* = 0.006, from 4.5% in 2009 to 1.0% in 2018, *p* < 0.001, and from 6.1% in 2009 to 3.6% in 2018, *p* < 0.001, respectively), there were no significant changes in the values of those aged younger than 1 years.

Table 2 shows the multivariate logistic regression analyses. The comparison of mortality risk in patients aged 13–17 years (comparative controls) with those in other age-groups did not show significant statistical differences. Of all patients, patients with an ISS ≥16 or survival probability ≥50%, or patients who underwent urgent blood transfusion, showed 3.24, 2.38, and 6.88 times higher odds of in-hospital mortality (*p* < 0.001). Moreover, in the total patients, patients with an ISS ≥16 or survival probability ≥50%, higher survival probability associated with lower odds of in-hospital mortality (*p* < 0.001, OR = 0.92, 95 % CI = 0.91–0.92; *p* < 0.001, OR = 0.93, 95% CI = 0.93–0.93; or *p* < 0.001, OR = 0.89, 95% CI = 0.88–0.90). When the year 2018 was used as a comparative control, patients with an ISS ≥ 16, and those with a survival probability ≥ 50% showed higher odds of in-hospital mortality before 2015.

## 4. Discussion

During the 10-year study period in Japan, the SMR and in-hospital mortality values among paediatric injured patients showed a significant yearly reduction. While they also showed significant yearly reductions among those with an ISS ≥16 and a survival probability ≥50%, no such reductions were observed among those <1 year old. Furthermore, multivariate logistic regression analyses indicated that patients who underwent urgent blood transfusion and patients with higher injury severity grade had a higher risk of in-hospital mortality. In addition, the mortality risk for paediatric injured patients showed a decreasing trend from 2009 to 2018.

Our study’s findings are in line with those of previous global studies that report yearly reductions in the in-hospital mortality among paediatric injured patients [1,4,5,6]. This study also showed that the in-hospital mortality risk in 2018 was one sixth that in 2009. Various factors that positively influence injury-related mortality have been reported over the past decade, such as social policy changes, the requirement of the fastening of seat belts in rear seats, the use of child seats for <6 years, the establishment of strengthened penalties for dangerous driving, and the use of safety devices with improved motor vehicle engineering; therefore, the in-hospital mortality improvements observed in this study can be attributed to multiple factors [1,2,11,12,17]. Our results show that the number of injured patients caused by traffic crashes and/or the proportion of those with polytrauma who had a higher mortality risk exhibited significant yearly in-hospital mortality reduction. Therefore, alterations in the mechanisms of injury due to social policy changes may have contributed to the in-hospital mortality decreases observed among our patients [18]. Next, the Joint Committee of the Japanese Association for the Surgery of Trauma developed a standardised injury care protocol and an off-the-job training course—JATEC in 2002, JPTEC in 2003, and Japan Advanced Trauma Evaluation and Care (JETEC) in 2014 [12,13,19]—in 2002 and established the nationwide JTDB [9]. Some nationwide studies using JTDB datasets had been performed to evaluate the quality of care and outcome values of injured patients, which might have contributed to the improvement of mortality in injured patients in Japan [10,11].

Previous studies [1,2,3,4,5,6] showed global differences in in-hospital mortality of paediatric injured patients, ranging from 0.12% to 8%. While the different survey methods make direct comparisons difficult, this study showed that the overall in-hospital mortality was 4.1%, not much lower than that of other developed countries [1,2,3,4,5,6], and that paediatric injured patients <1 year, with high TRISS Ps, and those who underwent urgent blood transfusion, had a high mortality risk, therefore, it is considered these factors will be most effective to improve injury care systems for decreasing the mortality rate of affected patients. In this study, the in-hospital mortality of those <1 year did not improve during the 10-year study period wherein no decreasing tendency was observed. A retrospective study in Australia showed that patient age <10 years, presence of traumatic head injury, and greater injury severity were associated with a higher mortality risk [2]. A previous nationwide Japanese analysis that focused on patients’ age-related characteristics showed that, among younger paediatric patients with severe injury, especially those <10 years, the proportion of those with traumatic head injury was higher than that of the other patients [16]. Furthermore, traumatic head injury is a leading cause of mortality and disability among children and adolescents [19,20,21,22]. Our multivariate logistic regression analysis also showed higher odds of in-hospital mortality among those with higher injury severity. Therefore, the establishment of an injury care system that provides high-quality care to younger paediatric patients with head injury and/or higher injury severity may improve the in-hospital mortality values of younger paediatric injured patients in Japan. There may be other factors in undiagnosed and unreported cases in clinical practice that potentially affected the mortality trend of younger paediatric injured patients. First, cases of intentional injury such as abusive head trauma may be included in this study cohort <1 year old. A previous study reported that several cases of child abuse existed among younger paediatric patients who had an injury event at home and that younger paediatric patients with abusive head trauma had higher mortality [23].

With regard to paediatric patients with high TRISS Ps, especially patients with cardiac arrest on hospital arrival, the incidence rate did not decrease over the study period in this study. A previous study showed that the rate of paediatric patients with traumatic out-of-hospital cardiac arrest (OHCA) who survived with good neurologic outcome was extremely low (3%) [24]. Therefore, injury prevention might be more effective to improve mortality and morbidity of paediatric injured patients with OHCA, along with the improvement of care for post-cardiac arrest.

Furthermore, this study showed that paediatric patients who underwent urgent blood transfusion had three times higher mortality risk than the others. Several studies suggested that damage control resuscitation including interventions such as early administration of blood products might be effective to reduce the mortality in paediatric patients with massive haemorrhage [25]. Providing high-quality injury care for paediatric patients with massive haemorrhage might help reduce the mortality of injured patients. Therefore, it is essential to continuously improve the quality of injury care systems based on the evaluation of not only outcome but also process indicators [26]. In Japan, there is as yet no complete case review system for injured patients based on JTDB data. To evaluate process indicators such as quality of prevention, diagnosis, and injury care, a nationwide review system for severely injured patients is needed.

Moreover, there is evidence of a volume–outcome relationship, wherein higher admission patient volumes and surgical volumes resulted in lower mortality per institution [27,28]. In this study, paediatric patients with severe injury, such as patients with OHCA and urgent blood transfusion, accounted for only 2% (*n* = 392) and 7% (*n* = 1201) respectively, of all paediatric injured patients during the 10-year study period. As the frequency of critical patients in our study was extremely low, appropriate similar high-quality injury care may not be provided to critical paediatric patients, compared to older patients. Future detailed studies such as full case review should focus on process clarification and outcomes influencing the prognosis of paediatric injured patients.

For a high-quality injury surveillance system able to accurately evaluate quality indicators of injury care such as the structure of the care system, care process, and injury outcomes, a dataset registering all inpatients in the survey target area is essential [1]. In a previous study using a dataset including all paediatric injured patients admitted to all the Australian hospitals [2], 686,409 paediatric injured patients aged <17 years old were examined during the 10-year study period from 2002 to 2012. On the other hand, the total number of paediatric injured patients <17 years old registered in JTDB from 2009 to 2018 was 25,028 (Figure 1). With regards to the paediatric population <15 years old in 2019, there were 15,744,000 Japanese children, which was three times that of Australian children (*n* = 4,920,000) [29]. That is to say, although Japan has three times the Australian paediatric population, this study population was unexpectedly low. It was suggested that the JTDB dataset may represent only part of the Japanese injured inpatients, because not all Japanese hospitals that treat injured patients participate in the JTDB, and also, the number of participating hospitals differed across the study period. In addition, 8108 (32% of the total) paediatric injured patients <17 years old registered in JTDB miss the date of injury severity and outcome. Considering this information regarding JTDB, there was large selection bias in our study, certainly an important limitation of our research. In the near future, there is a need for improved injury surveillance systems, with comprehensive datasets including all Japanese injured patients and minimum missing data, for the establishment of injury prevention strategies and the evaluation of the quality of injury care and related outcomes.

Next, burns are a noble cause of injury-related death and disability in younger children [1]. However, because the survival probability for burn cases was not calculated and input in the JTDB registry, we were unable to perform the additional analysis showing the SMR trend using TRISS Ps including burn cases. In a next research step, we should do analyses with all known injury death cases including burn cases to improve on preventive measures and injury care systems. Moreover, we did not conduct subgroup analyses based on the location of the injury event, type of hospital, or cost and quality of management. Several reports have shown that optimal treatment, according to the abovementioned factors, affects the outcomes. Therefore, despite the challenges associated with injury surveillance, future nationwide studies with subclass analyses should be conducted for the establishment of injury prevention strategies and improvements in the outcomes of paediatric injury patients. To the best of our knowledge, this study is the first to evaluate the in-hospital mortality trends of childhood injury in Japan, a country in which few injury surveillance systems aimed at improving the mortality of paediatric patients have been established so far.

## 5. Conclusions

The SMR and in-hospital mortality values of paediatric injured patients in Japan significantly decreased during the 10-year study period from 2009 to 2018. There is a need for improved injury surveillance systems for the establishment of injury prevention strategies along with the evaluation of the quality of injury care and outcome measures.

## Figures and Tables

**Figure 1 jcm-09-03273-f001:**
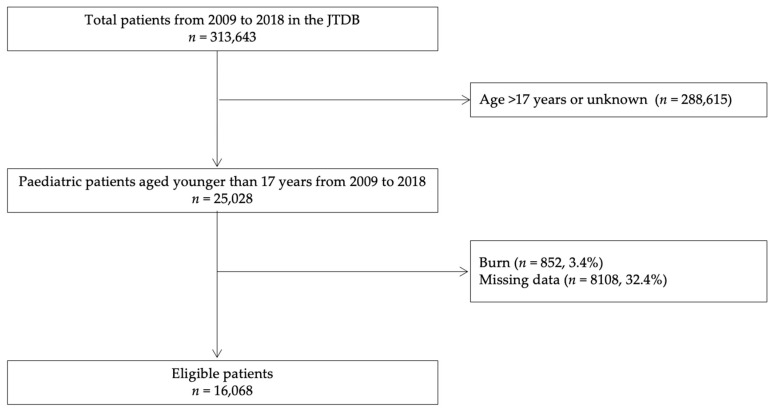
Flow diagram of study patient disposition. JTDB, Japanese Trauma Data Bank.

**Figure 2 jcm-09-03273-f002:**
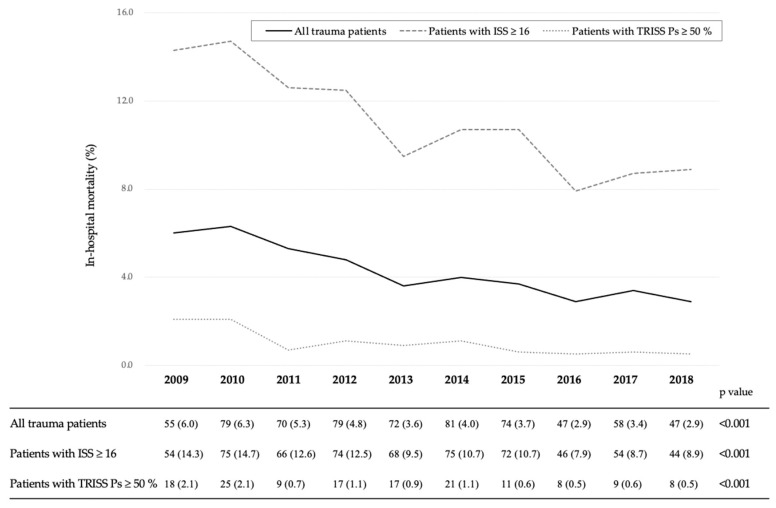
In-hospital mortality trends for paediatric injured patients by year. ISS, Injury Severity Score; TRISS, Trauma and Injury Severity Score; Ps, probability of survival.

**Figure 3 jcm-09-03273-f003:**
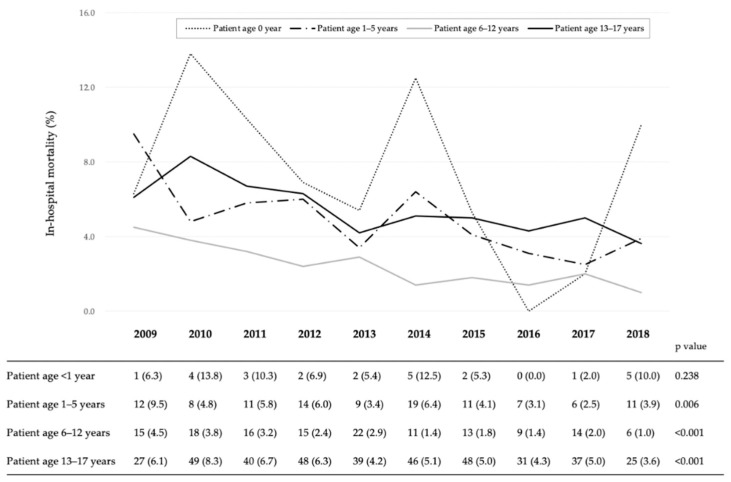
Yearly in-hospital mortality trends according to age group of paediatric injured patients.

**Table 1 jcm-09-03273-t001:** Demographics and other data of paediatric injured patients by year.

Variables	2009 *n* = 917	2010 *n* = 1258	2011 *n* = 1329	2012 *n* = 1637	2013 *n* = 1977	2014 *n* = 2023	2015 *n* = 1977	2016 *n* = 1614	2017 *n* = 1715	2018 *n* = 1621	*p* Value
Age in year, (median 5th–95th percentiles)	12 (2–17)	12 (2–17)	12 (2–17)	12 (2–17)	12 (2–17)	11 (2–17)	12 (2–17)	11 (2–17)	11 (1–17)	11 (1–15)	0.898
Patient age < 1 year, *n* (%)	16 (2)	29 (2)	29 (2)	29 (2)	37 (2)	40 (2)	38 (2)	29 (2)	49 (3)	50 (3)	0.085
Patient age 1–5 years, *n* (%)	127 (14)	166 (13)	191 (14)	234 (14)	262 (13)	298 (15)	271 (14)	227 (14)	240 (14)	285 (18)	0.030
Patient age 6–12 years, *n* (%)	331 (36)	471 (37)	508 (38)	614 (38)	759 (38)	781 (39)	708 (36)	632 (39)	689 (40)	600 (37)	0.260
Patient age 13–17 years, *n* (%)	443 (48)	592 (47)	601 (45)	760 (46)	919 (46)	904 (45)	960 (49)	726 (45)	737 (43)	686 (42)	0.003
Male, *n* (%)	653 (71)	907 (72)	957 (72)	1149 (70)	1380 (70)	1481 (73)	1437 (73)	1186 (73)	1236 (72)	1175 (72)	0.241
Transportation, *n* (%)
Transportation from the scene	697 (76)	978 (78)	1035 (78)	1266 (77)	1568 (79)	1636 (81)	1518 (77)	1238 (77)	1266 (74)	1175 (72)	<0.001
Transportation from another hospital	136 (15)	182 (14)	192 (14)	220 (13)	220 (11)	226 (11)	251 (13)	200 (12)	271 (16)	214 (13)	<0.001
Mechanism of injury, *n* (%)
Blunt	897 (98)	1245 (99)	1312 (99)	1608 (98)	1952 (99)	1999 (99)	1954 (99)	1593 (99)	1697 (99)	1609 (99)	0.101
Injury mechanism of blunt injury, *n* (%)
Road traffic crash	531 (59)	744 (60)	746 (57)	953 (59)	1127 (58)	1151 (58)	1081 (55)	837 (53)	846 (50)	728 (45)	<0.001
Fall	214 (24)	241 (19)	273 (21)	307 (19)	396 (20)	408 (20)	408 (21)	356 (22)	386 (23)	424 (26)	<0.001
Tumble	55 (6)	81 (7)	98 (7)	115 (7)	138 (7)	143 (7)	148 (8)	132 (8)	129 (8)	153 (10)	0.064
Cardiac arrest on hospital arrival, *n* (%)	23 (3)	38 (3)	36 (3)	42 (3)	44 (2)	44 (2)	51 (3)	33 (2)	40 (2)	41 (3)	0.864
Urgent blood transfusion, *n* (%)	83 (9)	109 (9)	111 (8)	140 (9)	151 (8)	140 (7)	130 (7)	111 (7)	117 (7)	109 (7)	0.055
Injury region, *n* (%)
Polytrauma	145 (16)	218 (17)	222 (17)	247 (15)	303 (15)	253 (13)	268 (14)	211 (13)	249 (15)	196 (12)	<0.001
Head injury with AIS ≥3	376 (41)	500 (40)	510 (38)	580 (35)	697 (35)	688 (34)	637 (32)	551 (34)	575 (34)	507 (31)	<0.001
Facial injury with AIS ≥3	7 (1)	11 (1)	12 (1)	14 (1)	18 (1)	14 (1)	26 (1)	19 (1)	22 (1)	11 (1)	0.442
Neck injury with AIS ≥3	2 (0.2)	2 (0.2)	4 (0.3)	5 (0.3)	1 (0.1)	4 (0.2)	3 (0.2)	4 (0.3)	2 (0.1)	2 (0.1)	0.774
Chest injury with AIS ≥3	188 (21)	265 (21)	267 (20)	308 (19)	391 (20)	363 (18)	257 (18)	281 (17)	304 (18)	264 (16)	0.017
Abdominal and pelvic injury with AIS ≥3	71 (8)	117 (9)	117 (9)	142 (9)	141 (7)	124 (6)	145 (7)	84 (5)	132 (8)	116 (7)	<0.001
Spinal injury with AIS ≥3	29 (3)	40 (3)	52 (4)	60 (4)	69 (3)	73 (4)	76 (4)	68 (4)	82 (5)	64 (4)	0.482
Upper extremity injury with AIS ≥3	81 (9)	135 (11)	151 (11)	144 (9)	154 (8)	155 (8)	171 (9)	202 (13)	173 (10)	190 (12)	<0.001
Lower extremity injury with AIS ≥3	152 (17)	198 (16)	197 (15)	237 (14)	293 (15)	311 (15)	292 (15)	214 (13)	259 (15)	248 (15)	0.636
Injury Severity Score, (median 5–95th percentiles)	10 (1–34)	10 (1–36)	10 (1–36)	10 (1–35)	10 (1–35)	9 (1–35)	9 (1–34)	10 (1–34)	9 (1–33)	9 (1–33)	<0.001
Injury Severity Score ≥16 *n* (%)	377 (71)	511 (71)	523 (72)	591 (72)	717 (71)	698 (71)	671 (70)	579 (72)	624 (75)	496 (68)	0.246
Revised Trauma Score, (median 5–95th percentiles)	7.84 (4.09–7.84)	7.84 (4.09–7.84)	7.84 (4.09–7.84)	7.84 (4.09–7.84)	7.84 (4.42–7.84)	7.84 (4.80–7.84)	7.84 (4.91–7.84)	7.84 (5.03–7.84)	7.84 (4.21–7.84)	7.84 (4.91–7.84)	<0.001
TRISS Ps, (median 5–95th percentiles)	99.1 (59.1–99.6)	99.2 (50.5–99.7)	99.2 (42.6–99.7)	99.2 (97.8–99.6)	99.3 (55.1–99.7)	99.3 (66.6–99.7)	99.3 (72.1–99.7)	99.4 (72.3–99.7)	99.3 (72.1–99.7)	99.3 (68.4–99.7)	<0.001
Patients with TRISS Ps ≥50%, *n* (%)	873 (95)	1196 (95)	1259 (95)	1561 (95)	1904 (96)	1946 (96)	1903 (96)	1558 (97)	1646 (96)	1561 (96)	0.148
Actual in-hospital mortality, *n* (%)	55 (6.0)	79 (6.3)	70 (5.3)	79 (4.8)	72 (3.6)	81 (4.0)	74 (3.7)	47 (2.9)	58 (3.4)	47 (2.9)	<0.001
Standardised mortality ratio	0.88	0.93	0.74	0.76	0.63	0.74	0.68	0.57	0.61	0.58	0.006

AIS, Abbreviated Injury Scale; TRISS, Trauma and Injury Severity Score; Ps, probability of survival.

**Table 2 jcm-09-03273-t002:** Multivariate logistic regression analysis of in-hospital mortality among paediatric injured patients.

	All Patients *n* = 16,068			Patients with ISS ≥16 *n* = 5787			Patients with TRISS ≥50% *n* = 15,407		
	OR	(95% CI)	*p* Value	OR	(95% CI)	*p* Value	OR	(95% CI)	*p* Value
Age in years									
<1 year	1.75	(0.83 to 3.67)	0.140	1.58	(0.75 to 3.35)	0.233	2.21	(0.87 to 5.60)	0.096
1–5 years	1.00	(0.81 to 1.24)	0.992	0.98	(0.79 to 1.22)	0.845	1.16	(0.89 to 1.52)	0.274
6–12 years	0.89	(0.79 to 1.00)	0.054	0.90	(0.80 to 1.02)	0.102	0.93	(0.80 to 1.09)	0.363
13–17 years	1.00	−		1.00	−		1.00	−	
Urgent blood transfusion	3.24	(2.38 to 4.41)	<0.001	2.38	(1.75 to 3.22)	<0.001	6.88	(4.62 to 10.25)	<0.001
TRISS Ps	0.92	(0.91 to 0.92)	<0.001	0.93	(0.93 to 0.93)	<0.001	0.89	(0.88 to 0.90)	<0.001
Year									
2009	6.16	(2.94 to 12.88)	<0.001	5.90	(2.78 to 12.56)	<0.001	5.35	(2.05 to 13.97)	0.001
2010	6.53	(3.26 to 13.08)	<0.001	5.88	(2.88 to 12.00)	<0.001	5.91	(2.37 to 14.78)	<0.001
2011	2.49	(1.18 to 5.23)	0.016	2.40	(1.14 to 5.05)	0.022	1.54	(0.53 to 4.43)	0.424
2012	2.79	(1.39 to 5.57)	0.004	2.89	(1.43 to 5.85)	0.003	1.91	(0.73 to 5.00)	0.185
2013	2.03	(1.03 to 4.00)	0.042	2.10	(1.05 to 4.19)	0.035	1.24	(0.47 to 3.24)	0.667
2014	2.97	(1.51 to 5.86)	0.002	2.98	(1.49 to 5.96)	0.002	2.59	(1.02 to 6.58)	0.046
2015	2.94	(1.01 to 4.12)	0.047	2.12	(1.05 to 4.32)	0.037	1.13	(0.40 to 3.14)	0.820
2016	1.18	(0.55 to 2.54)	0.665	1.27	(0.60 to 2.72)	0.532	0.92	(0.30 to 2.80)	0.887
2017	1.31	(0.63 to 2.73)	0.465	1.25	(0.60 to 2.61)	0.556	0.92	(0.31 to 2.74)	0.887
2018	1.00	−		1.00	−		1.00	−	

OR, odds ratio; CI, confidence interval; TRISS, Trauma and Injury Severity Score; Ps, probability of survival.

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
