# Peer review of "Ten-Year in-Hospital Mortality Trends among Paediatric Injured Patients in Japan: A Nationwide Observational Study"

_jcm, 2020, doi:10.3390/jcm9103273_

Round 1

Reviewer 1 Report

GENERAL COMMENTS:

Thank you for the opportunity to review this paper, which seeks to document and discuss changes in paediatric (<17y) trauma-related mortality in the Japanese population. The authors have analysed data from the Japanese Trauma Data Base, and report a noteworthy reduction in paediatric trauma-related mortality of the 10-year period 2009-2018. This reduction corresponds with a similarly reported reduction adult (17+y) trauma-related mortality from the same data source. The reduced mortality is attributed to be likely successes of preventive measures and improved trauma systems. 

All these are laudable observations and many will welcome the further examples of trauma prevention and better trauma management. However, irrespective of their laudable nature, I am concerned that the data source, dataset and analysis presented in this project are not yet sufficiently declared and defined to allow Reviewers and Readers alike to be convinced that these observations and attributions are indeed valid. The following review will highlight these and other concerns, which form the basis of my current conclusion that the manuscript not be accepted for publication. However, it is possible that with major revision the manuscript may well be suitable for future publication. 

ABSTRACT:

I would recommend the Authors soften the language of the final and concluding sentences. At best, the data presented here may be explained in terms of successful preventive measures and/or improved trauma systems. However, I do not feel the Authors present data or discussion to justify a statement as dogmatic as, "The provision of high-quality care to younger paediatric patients or/and severely injured patients yield reduced in-hospital mortality values." This sentence should be re-worded, please.

INTRODUCTION:

(a) "Unintentional injury"?

The Authors introduce the significant global burden of death and disability due to childhood injury. As is detailed further in the comments about 'language' below, I caution the authors on their use of "unintentional injury" in this Introduction text. "Intentional" injuries are also important causes of trauma in childhood, making the opening statements of the introduction unhelpfully exclusive of injuries causes such as interpersonal and self-directed violence.

(b) Is childhood injury continuing to decline?

The Authors introduce data to support decreasing rates of injury in other populations, with sentence and citations, "However, its incidence rate was found to have decreased from 18.2 % in 1960 to 4.3 % in 2015 through the implementation of preventive measures and development of appropriate trauma care systems [1–4]." Assuming that "its" refers here to "non-intentional trauma", I cannot agree with this statement in terms of the citations provided. Indeed, one of these citations refers to an Australian study, which to my reading reports no reduction in childhood injury over a similar 10-year period to that being reported by the current manuscript. The important recognition by these Australian authors (Mitchell et al, 2018) that sustained commitment to public health and other preventative measures had no resulted in reduction in childhood injury rates speaks to the very complexity of this topic. Unfortunately, this complexity is seemingly overlooked by the Authors in this Introduction. 

Given that it follows on from a discussion of data spanning 1960's to 2015, the cited CDC annual figure of injury needs to be given a year timeframe please. It is published in 2018, but what year or years does it report on?

(c) Justification of hypothesis/aim

The Discussion informs us that important changes were made to improve the standardisation of trauma care and education in 2002, including "a standardised trauma care protocol and an off-the job training course—Japan Advanced Trauma Evaluation and Care—in 2002 and established the nationwide JTDB." If these changes were implemented in 2002, and cited Japanese department of health figures indicate no reduction in trauma-related in-hospital mortality between 2004-06 and 2009-11 (refs #4 and #5), why would these measures bring about reductions in mortality only a decade later, i.e. in the period post 2012? Were there additional developments to suggest this timeframe was relevant, or is this based solely on the findings of the adult cohort analysis. If so, both analyses may be subject to similar biases, predisposing to affirming, but perhaps not representative (or "correct") findings. This hypothesis needs further development, please.

METHODOLOGY:

(a) Description of the Japan Trauma Data Base (JTDB)

Please can the Authors provide more information regarding the Japan Trauma Data Base (JTDB). More concern is this trauma registry may not reflect a national dataset in the way the Title and Manuscript text might easily imply. I may also be incorrect about this, but I feel Readers need to know more about which hospitals (in terms of number, character, coverage) contribute to this data base. How many hospitals? Are these all major trauma centres, or a mixture of non-major and major trauma centres? What proportion of Japan's trauma receiving hospital's does this represent?

My concerns that this data base may represent only part of the Japanese trauma data set is raised by the numbers of reported paediatric cases as per Figure 1: 25,028 all cases <17y of which 14,324 had ISS of 3+. Even with this very low ISS threshold for inclusion, the total injury numbers for a population the size of Japan is remarkably low.

By way of counter example, the paper by Mitchell et al. (2018) cited here as reference #4 examined an Australian paediatric cohort of similar age and timeframe. Whilst the Australian authors have used ICISS to categorise their trauma cohort into mild, moderate and serious groups - thus making direct comparison with this ISS 3+ cohort difficult - the differences in numbers are striking. Australia has a total population one fifth the size of Japan, and yet the Australian 10-year study identified >47,000 injured children <17y in their "serious" (ICISS<0.942) group alone, with an additional >218,000 injured children in the "moderate" (ICISS 0.942–0.99) group.

I consider it essential for the Authors to better define the epidemiology of their own 14,000 strong cohort, to make sense of this unexpectedly low number. I would also consider this a point of central discussion when the wider applications and/or limitations of the study findings are being developed in the Discussion.

(b) Exclusion criteria

I would like to raise concerns about the exclusion criteria. I concede the Authors have commented these exclusion criteria were designed to match those of a previous study examining adult trauma mortality, but I do not agree that this justification is sufficient to overlook known cohorts of trauma-related deaths.

(i) Cardiac arrest

Most importantly, I disagree strongly with the decision to exclude trauma cases who present in cardiac arrest. I acknowledge similar examples of such exclusion can be found in the literature. However, it is correct to say that these children die/are declared dead "in hospital", and so many trauma systems regard cases of this nature, however futile, as "in-hospital" deaths, and this too is supported by reported series inclusive of cardiac arrest trauma cases.

With apologies for any incorrect calculations on my part, I would suggest that had trauma cardiac cases been included, that would have represented about 3.3% of the total study cohort. This is an important an important number, exceeding in total the otherwise reported death cohort of 315/14324. As such, assuming all of these cases died which is likely but not a given, the overall mortality wound have more than doubled, rising from 2.2% (315/14342) to 5.5% (816/14843).

(ii) Burns

Also, I would challenge the Authors decision to exclude burns cases. Burns is a very important injury cause of death and disablity in childhood. Burns management, like other areas of trauma, benefits from a prevention as well as systems-driven approach to pre-hospital and in-hospital care... all of which may be drivers for observed changes in in-hospital outcome.

To ignore known cases of trauma-related mortality, is to ignore opportunities to learn, mature and reduce the burden of injury within a population. This is especially true of a manuscript, which goes on to attribute reductions in mortality to improved preventive measures and systems of trauma care. Whilst improvements in in-hospital care will struggle to impact the outcomes of trauma cases presenting in cardiac arrest, there are very real and important opportunities for preventive measures and pre-hospital systems of trauma care to do so. Therefore, please can I strongly encourage the Authors to repeat their analyses with all known trauma death cases, including those presenting in cardiac arrest and burns. If the same trends for decline are evident, this would provide a far stronger evidence base for development in the discussion.

RESULTS:

Please consider expanding Table 1 to include a figure for the number of deaths reported for each year. I am aware that these numbers can be arrived at by working backwards from some of the percentages reported in Figure 2 and 3, but Readers should not be left trying to derive data of such central importance to the project being reported. Indeed, the I would also recommend expanding the description of the death rates in the text, which at present is only briefly introduced in the first paragraph of the results section as, "The overall in-hospital mortality was 2.2 %, that of patients with an ISS ≥ 16 was 5.5 %, and that of patients with a survival probability ≥ 50 % was 1.0%. The overall SMR was 0.54. Tables 1 ..." Given the multiple references to "N=" in the previous sentences, it seems unnecessary to skip so quickly over data of central importance to a paper on mortality rates.

Please consider writing a more detailed legend for Figure 2, to increase clarity about the numbers in the table representing in-hospital death rates. Would the authors consider presenting death rates in both whole numbers and percentages. I feel both are important aspects of any trauma series... every death counts. It also allows Readers to make greater sense of the trends being reported year on year.

DISCUSSION:

(a) Critical appraisal

My assessment is the Discussion stop short of a critical appraisal of this study and the surrounding literature. Only 9 new references are introduced in the Discussion and the 10 or so references cited in the Introduction are not considered further in critical discussion of the reported findings. There is more written about paediatric trauma mortality, trends in trauma mortality, risks of death in trauma mortality, and the limitations of looking only at in-hospital mortality to the exclusion of out-of-hospital trauma mortality.

(b) Justification of discussion and conclusions

The findings are a little too readily attributed to the hypothesised impact of preventive measures and improved trauma systems, but this aspect of the discussion lacks the necessary detail or deliberation to justify this over and above... "i guess it makes sense". I do not challenge the Authors to denigrate the Authors, their project or indeed their assertion that preventive measures and improved trauma systems may underly reductions in paediatric in-hospital trauma mortality. Rather, I am concerned the Authors have stopped too short in their discussion, and so not provided the necessary scholarly justification for me and other Readers to simply accept these assertions.

c) Limitations and biases

I commend the Authors for their consideration of limitations, and even this can be expanded. I would encourage the Authors to bring more of this rigor and reflection into the earlier parts of the Discussion. What role are reporting and selection biases playing here?

For example: given the variation in hospitals reporting to the JTDB over the course of the study, is it possible that the observed reduction in trauma in-hospital mortality is not real? What considerations allow the Authors to assert that these are real reductions?

For example: how do the authors reconcile no reported absolute reduction (indeed some increase at times) in registry trauma cases over the study time period, with the observed reduction in in-hospital mortality? Are the preventive measures and improved trauma systems reducing the lethality of trauma whilst not decreasing injury per se? Again, could this reflect changes in which and how many centres reported over time?

(c) Agreed need to improved surveillance and reporting mechanisms

I completely support the Authors comments that there is a need for improved surveillance and reporting mechanisms devoted to childhood injury. It is for this reason that I would encourage a more "warts and all" reporting of the Authors methods and findings, as this will support their claim that they have been left needing to make many assumptions because of the limitations of the data available, and yet have findings that would encourage these surveillance and reporting mechanisms to be developed and used.

COMMENTS ABOUT LANGUAGE:

In addition to the concerns, the overall language of the manuscript warrants improvement. I make these comments cognisant that the Authors likely do not speak English as a first language, and I would congratulate them on the overall quality of the written language. 

Most importantly, the Authors can improve the trauma-specific language in this manuscript. As is well outlined in the World Health Organisation document available at https://www.who.int/ceh/capacity/injuries.pdf?ua=1 and elsewhere, English language trauma literature has stepped intentionally AWAY from the use of "accident" to describe "trauma events". The shared concern is that the term "accident" perpetuates a societal (and scholarly) misperception that trauma events are inevitable... "accidents just happen"). Instead, trauma literature should assert in language and message the important truth that the vast majority of trauma is PREVENTABLE. Therefore I would strongly encourage the Authors to use terms such as "road traffic crashes" instead of "road traffic accidents" and "trauma events" rather than "accidents".

I am also unsure whether the Authors defined trauma as "non-intentional injury" with the intent (no pun intended) to exclude intentional injury. Irrespective of whether we are considering paediatric trauma more generally, or paediatric trauma-related deaths more specifically, the tragic contributions of abusive head trauma, other forms of inflicted injury, self-harm and suicide should not be excluded from this research and discussion. From my reading

There are also several jarring grammatical errors, which can be screened out by the Authors with or without the assistance of a service specialising in scientific English writing. For example, the use of plural vs singular. The Authors should also beware use of long and convoluted sentences with many subordinate clauses.

FINAL COMMENTS:

I would like to encourage the Authors to consider this and other reviews as an opportunity to refine this manuscript into a form that will permit publication, as the message is an important one. I feel these refinements stand in the way of your project and publication making the impact you have all worked hard to achieve.

Thank you again for the opportunity to review this process.

Author Response

Reviewer’s comments (in blue) and Answers (in black)

We want to thank the reviewers for their insightful comments on our paper. We feel the comments have helped us significantly improve the manuscript.

Reviewer: 1

Major comments.

ABSTRACT

  1. I would recommend the Authors soften the language of the final and concluding sentences. At best, the data presented here may be explained in terms of successful preventive measures and/or improved trauma systems. However, I do not feel the Authors present data or discussion to justify a statement as dogmatic as, "The provision of high-quality care to younger paediatric patients or/and severely injured patients yield reduced in-hospital mortality values." This sentence should be re-worded, please.

Response: We agree that concluding sentence in the abstract section was formulated without scientific evidence. We have deleted this sentence and addressed the challenges for the future as follows:

Based on our results, there is a need for improved injury surveillance systems for the establishment of injury prevention strategies along with evaluation of the quality of injury care and outcome. 

INTRODUCTION

  1. (a) "Unintentional injury"?

The Authors introduce the significant global burden of death and disability due to childhood injury. As is detailed further in the comments about 'language' below, I caution the authors on their use of "unintentional injury" in this Introduction text. "Intentional" injuries are also important causes of trauma in childhood, making the opening statements of the introduction unhelpfully exclusive of injuries causes such as interpersonal and self-directed violence.

Response: Thank you for your valuable comments. We have characterised “injury” as either non-intentional or intentional. Therefore, we have deleted unintentional injury in the introduction section.

  1. (b) Is childhood injury continuing to decline?

The Authors introduce data to support decreasing rates of injury in other populations, with sentence and citations, "However, its incidence rate was found to have decreased from 18.2 % in 1960 to 4.3 % in 2015 through the implementation of preventive measures and development of appropriate trauma care systems [1–4]." Assuming that "its" refers here to "non-intentional trauma", I cannot agree with this statement in terms of the citations provided. Indeed, one of these citations refers to an Australian study, which to my reading reports no reduction in childhood injury over a similar 10-year period to that being reported by the current manuscript. The important recognition by these Australian authors (Mitchell et al, 2018) that sustained commitment to public health and other preventative measures had no resulted in reduction in childhood injury rates speaks to the very complexity of this topic. Unfortunately, this complexity is seemingly overlooked by the Authors in this Introduction.

Given that it follows on from a discussion of data spanning 1960's to 2015, the cited CDC annual figure of injury needs to be given a year timeframe please. It is published in 2018, but what year or years does it report on?

Response: We agree with your suggestions. We have revised the structure of the Introduction section, based on the newly referenced papers.

  1. (c) Justification of hypothesis/aim

The Discussion informs us that important changes were made to improve the standardisation of trauma care and education in 2002, including "a standardised trauma care protocol and an off-the job training course—Japan Advanced Trauma Evaluation and Care—in 2002 and established the nationwide JTDB." If these changes were implemented in 2002, and cited Japanese department of health figures indicate no reduction in trauma-related in- hospital mortality between 2004-06 and 2009-11 (refs #4 and #5), why would these measures bring about reductions in mortality only a decade later, i.e. in the period post 2012? Were there additional developments to suggest this timeframe was relevant, or is this based solely on the findings of the adult cohort analysis. If so, both analyses may be subject to similar biases, predisposing to affirming, but perhaps not representative (or "correct") findings. This hypothesis needs further development, please.

Response: Thank you for your valuable comments. Indeed, we did not show the scientific evidence supporting whether the off-the job training and guidelines may lead to reductions in in-hospital mortality. Therefore, we have deleted the hypothesis shown in the pre-revision manuscript.

METHODOLOGY

  1. (a) Description of the Japan Trauma Data Base (JTDB)

Please can the Authors provide more information regarding the Japan Trauma Data Base (JTDB). More concern is this trauma registry may not reflect a national dataset in the way the Title and Manuscript text might easily imply. I may also be incorrect about this, but I feel Readers need to know more about which hospitals (in terms of number, character, coverage) contribute to this data base. How many hospitals? Are these all major trauma centres, or a mixture of non-major and major trauma centres? What proportion of Japan's trauma receiving hospital's does this represent?

My concerns that this data base may represent only part of the Japanese trauma data set is raised by the numbers of reported paediatric cases as per Figure 1: 25,028 all cases <17y of which 14,324 had ISS of 3+. Even with this very low ISS threshold for inclusion, the total injury numbers for a population the size of Japan is remarkably low.

By way of counter example, the paper by Mitchell et al. (2018) cited here as reference #4 examined an Australian paediatric cohort of similar age and timeframe. Whilst the Australian authors have used ICISS to categorise their trauma cohort into mild, moderate and serious groups - thus making direct comparison with this ISS 3+ cohort difficult - the differences in numbers are striking. Australia has a total population one fifth the size of Japan, and yet the Australian 10-year study identified >47,000 injured children <17y in their "serious" (ICISS<0.942) group alone, with an additional >218,000 injured children in the "moderate" (ICISS 0.942–0.99) group.

I consider it essential for the Authors to better define the epidemiology of their own 14,000 strong cohort, to make sense of this unexpectedly low number. I would also consider this a point of central discussion when the wider applications and/or limitations of the study findings are being developed in the Discussion.

Response: Accordingly, we have added more information regarding participating hospitals of JTDB in the Methods section as follows:

JTDB data collection started from 55 hospitals in 2003. The number of participating hospitals increased year by year, up to a total of 280 hospitals in all 47 prefectures in Japan, including 92% of government-approved tertiary emergency medical centres in March, 2019.

We agree with your concern that the JTDB dataset include information on only part of all injured inpatients in Japan. Moreover, as you pointed out, this concern about the unexpectedly low number of paediatric patients registered in JTDB was an important discussion point for improvement of the injury surveillance systems and establishment of injury prevention strategies. To make this point clearer, we have added the following to the discussion section as limitation:

       For a high-quality injury surveillance system able to accurately evaluate quality indicators of injury care such as structure of care system, care process, and injury outcome, a dataset registering all inpatients in the survey target area is essential [1]. In previous study using a dataset including all paediatric injured patients admited to all the hospital in Australia [2], 6 86 409 paediatric injured patients < 17 years old were examined during the 10-years study period from 2002 to 2012. On the other hand, the total number of paediatric injured patients <17 years old registered in JTDB from 2009 to 2018 was 25 028 (Figure 1). With regards to the paediatric population <15 years old in 2019, there were 15 744 000 Japanese children, which was three times that of Australian children (N = 4 920 000) [29]. That is to say, although Japanese has three times the Australian paediatric population, this study population was unexpected low. It was suggested that JTDB dataset may represent only part of the Japanese injured inpatients, because not all Japanese hospital that treat injured patients participate in the JTDB, and also, the number of participating hospitals differed across the study period. In addition, 8108 (32 % of total) paediatric injured patients <17 years old registered in JTDB miss the date of injury severity and outcome. Considering this information regarding JTDB, there was large selection bias in our study, certainly an important limitation of our research. In the near future, there is a need for improved injury surveillance systems, with comprehensive datasets including all Japanese injured patients and minimum missing data, for the establishment of injury prevention strategies and the evaluation of the quality of injury care and related outcomes.

  1. (b) Exclusion criteria

I would like to raise concerns about the exclusion criteria. I concede the Authors have commented these exclusion criteria were designed to match those of a previous study examining adult trauma mortality, but I do not agree that this justification is sufficient to overlook known cohorts of trauma-related deaths.

(i) Cardiac arrest

Most importantly, I disagree strongly with the decision to exclude trauma cases who present in cardiac arrest. I acknowledge similar examples of such exclusion can be found in the literature. However, it is correct to say that these children die/are declared dead "in hospital", and so many trauma systems regard cases of this nature, however futile, as "in-hospital" deaths, and this too is supported by reported series inclusive of cardiac arrest trauma cases.

With apologies for any incorrect calculations on my part, I would suggest that had trauma cardiac cases been included, that would have represented about 3.3% of the total study cohort. This is an important an important number, exceeding in total the otherwise reported death cohort of 315/14324. As such, assuming all of these cases died which is likely but not a given, the overall mortality wound have more than doubled, rising from 2.2% (315/14342) to 5.5% (816/14843).

Response: Thank you for your valuable comment. We strongly agree with you, and have therefore edited the Methods and Results of the additional work suggested by the reviewers. We have performed again statistical analysis for paediatric injured patients including cardiac arrest on arrival (N = 392, 2.4% of total).

  1. (ii) Burns

Also, I would challenge the Authors decision to exclude burns cases. Burns is a very important injury cause of death and disablity in childhood. Burns management, like other areas of trauma, benefits from a prevention as well as systems-driven approach to pre-hospital and in-hospital care... all of which may be drivers for observed changes in in-hospital outcome.

To ignore known cases of trauma-related mortality, is to ignore opportunities to learn, mature and reduce the burden of injury within a population. This is especially true of a manuscript, which goes on to attribute reductions in mortality to improved preventive measures and systems of trauma care. Whilst improvements in in-hospital care will struggle to impact the outcomes of trauma cases presenting in cardiac arrest, there are very real and important opportunities for preventive measures and pre- hospital systems of trauma care to do so. Therefore, please can I strongly encourage the Authors to repeat their analyses with all known trauma death cases, including those presenting in cardiac arrest and burns. If the same trends for decline are evident, this would provide a far stronger evidence base for development in the discussion.

Response: We agree on the worth of excluding burn cases from the objects of study. However, because survival probability for burn cases was not calculated and input in JTDB registry, we are unable to perform this additional analysis including burn cases. Therefore, we have modified the reason why burn cases were excluded as the limitation of our study as follows:

Next, burn are an important cause of injury-related death and disability in younger children [1]. However, because the survival probability for burn case was not calculated and input in the JTDB registry, we were unable to perform the additional analysis showing the SMR trend using TRISS Ps including burn cases. In the next research step, we should do analyses with all known injury death cases including burn cases to improve on preventive measures and injury care system.

RESULTS

  1. Please consider expanding Table 1 to include a figure for the number of deaths reported for each year. I am aware that these numbers can be arrived at by working backwards from some of the percentages reported in Figure 2 and 3, but Readers should not be left trying to derive data of such central importance to the project being reported. Indeed, the I would also recommend expanding the description of the death rates in the text, which at present is only briefly introduced in the first paragraph of the results section as, "The overall in-hospital mortality was 2.2 %, that of patients with an ISS ≥ 16 was 5.5 %, and that of patients with a survival probability ≥ 50 % was 1.0%. The overall SMR was 0.54. Tables 1 ..." Given the multiple references to "N=" in the previous sentences, it seems unnecessary to skip so quickly over data of central importance to a paper on mortality rates.

Please consider writing a more detailed legend for Figure 2, to increase clarity about the numbers in the table representing in-hospital death rates. Would the authors consider presenting death rates in both whole numbers and percentages. I feel both are important aspects of any trauma series... every death counts. It also allows Readers to make greater sense of the trends being reported year on year.

Response: Accordingly, we have revised the descriptions about the number of deaths in the main text, Table 1, and Figures 1 and 2. Moreover, we have added a detailed legend for Figures 2 and 3.

DISCUSSION:

  1. (a) Critical appraisal

My assessment is the Discussion stop short of a critical appraisal of this study and the surrounding literature. Only 9 new references are introduced in the Discussion and the 10 or so references cited in the Introduction are not considered further in critical discussion of the reported findings. There is more written about paediatric trauma mortality, trends in trauma mortality, risks of death in trauma mortality, and the limitations of looking only at in- hospital mortality to the exclusion of out-of-hospital trauma mortality.

Response: Thank you for your valuable comments. We have added more information on mortality, mortality trend, and mortality risk in the Discussion sections.

  1. (b) Justification of discussion and conclusions

The findings are a little too readily attributed to the hypothesised impact of preventive measures and improved trauma systems, but this aspect of the discussion lacks the necessary detail or deliberation to justify this over and above... "i guess it makes sense". I do not challenge the Authors to denigrate the Authors, their project or indeed their assertion that preventive measures and improved trauma systems may underly reductions in paediatric in- hospital trauma mortality. Rather, I am concerned the Authors have stopped too short in their discussion, and so not provided the necessary scholarly justification for me and other Readers to simply accept these assertions.

Response: Thank you for this pertinent suggestion. Accordingly, we have revised the Discussion to reflect more on scientific-based evidences supporting our data.

  1. c) Limitations and biases

I commend the Authors for their consideration of limitations, and even this can be expanded. I would encourage the Authors to bring more of this rigor and reflection into the earlier parts of the Discussion. What role are reporting and selection biases playing here?

For example: given the variation in hospitals reporting to the JTDB over the course of the study, is it possible that the observed reduction in trauma in-hospital mortality is not real? What considerations allow the Authors to assert that these are real reductions?

For example: how do the authors reconcile no reported absolute reduction (indeed some increase at times) in registry trauma cases over the study time period, with the observed reduction in in-hospital mortality? Are the preventive measures and improved trauma systems reducing the lethality of trauma whilst not decreasing injury per se? Again, could this reflect changes in which and how many centres reported over time?

Response: Thank you for your valuable comments. Accordingly, we have added the limitations of this study such as selection bias,… to the Discussion.

  1. (c) Agreed need to improved surveillance and reporting mechanisms

I completely support the Authors comments that there is a need for improved surveillance and reporting mechanisms devoted to childhood injury. It is for this reason that I would encourage a more "warts and all" reporting of the Authors methods and findings, as this will support their claim that they have been left needing to make many assumptions because of the limitations of the data available, and yet have findings that would encourage these surveillance and reporting mechanisms to be developed and used.

Response: We agree that the Japanese injury surveillance system is insufficient that there is a need for improvement. To make this point clear, we have added this point to the Discussion and Conclusion sections.

COMMENTS ABOUT LANGUAGE

  1. In addition to the concerns, the overall language of the manuscript warrants improvement. I make these comments cognisant that the Authors likely do not speak English as a first language, and I would congratulate them on the overall quality of the written language.

Most importantly, the Authors can improve the trauma- specific language in this manuscript. As is well outlined in the World Health Organisation document available at https://www.who.int/ceh/capacity/injuries.pdf?ua=1 and elsewhere, English language trauma literature has stepped intentionally AWAY from the use of "accident" to describe "trauma events". The shared concern is that the term "accident" perpetuates a societal (and scholarly) misperception that trauma events are inevitable... "accidents just happen"). Instead, trauma literature should assert in language and message the important truth that the vast majority of trauma is PREVENTABLE. Therefore I would strongly encourage the Authors to use terms such as "road traffic crashes" instead of "road traffic accidents" and "trauma events" rather than "accidents".

I am also unsure whether the Authors defined trauma as "non-intentional injury" with the intent (no pun intended) to exclude intentional injury. Irrespective of whether we are considering paediatric trauma more generally, or paediatric trauma-related deaths more specifically, the tragic contributions of abusive head trauma, other forms of inflicted injury, self-harm and suicide should not be excluded from this research and discussion. From my reading

There are also several jarring grammatical errors, which can be screened out by the Authors with or without the assistance of a service specialising in scientific English writing. For example, the use of plural vs singular. The Authors should also beware use of long and convoluted sentences with many subordinate clauses.

Response: Thank you for your valuable comments. As you pointed out, we used terms such as accident, trauma, or injury without precisedefinition. Therefore, we have characterised “injury” as either non-intentional or intentional. Therefore, we have changed from trauma to injury and accident to event. Finally, we have had the manuscript revised by an experienced scientific editor, who improved on the grammar and style of the manuscript.

FINAL COMMENTS:

  1. I would like to encourage the Authors to consider this and other reviews as an opportunity to refine this manuscript into a form that will permit publication, as the message is Submission Data of this review an important one. I feel these refinements stand in the way of your project and publication making the impact you have all worked hard to achieve.

Response: Again, thank you for giving us the opportunity to strengthen our manuscript with your valuable comments and queries. We have worked hard to incorporate your feedback and hope that this revision meets your requirements for high quality in publication.

Reviewer 2 Report

The authors describe a large epidemiologic study of pediatric trauma mortality over a ten-year period in Japan. The objectives are clear, the data are well-presented and the conclusions are justified. I have a few minor comments that may improve the quality of the paper:

  1. P2 L51: Do the authors mean “estimation of the effectiveness of injury prevention measures” instead of “estimation of injury prevention measures”?
  2. P2 L72: The authors indicate that they included only patients with ISS >= 3 in accordance with a previous paper. Can they clarify the rationale for this threshold in a few words in this paragraph?
  3. The authors excluded patients who were dead on arrival, which is somewhat justifiable since they are looking at in-hospital mortality. However, I think it would be useful to show the trend in pre-hospital mortality over time maybe in a supplementary table or figure. It is possible that in-hospital mortality is decreasing at the same time that pre-hospital mortality is increasing? It would be worthwhile proving or disproving this with their data.
  4. Likewise, the authors excluded patients with missing data. What was the pattern of missing data? Did the amount of missing data change over the 10-year period? If it did, e.g. if by the end of the period the TRISS could be calculated on a higher proportion of patients than at the beginning, this may skew mortality estimates (ascertainment bias).
  5. P3 L96-98: I do not understand this sentence. It should be reworded, e.g. “the Cochran-Armitage test was used to test for trends in SMR and in-hospital mortality by study year, which was treated as an ordinal variable.”
  6. The authors presented the median and IQR for continuous data. My personal preference is to use the median and 5th – 95th percentiles since they give a better indication of the population distribution than the IQR.
  7. P4 L104: What does “mean … for categorical variables” mean?
  8. P4 L115-117: Why were these age thresholds selected? I think the authors should rename “0 years” as “<1 year”.
  9. The authors should present transfusion data, If available, in Table 1 since the requirement for blood products is an independent predictor of trauma mortality. Has blood product utilization changed in this patient population over the ten-year period?
  10. The authors have lost some of the granularity of their data by using only the year of death in the regression model. It would be interesting to see a time series analysis of the data using day or month of death, and then assessing for any trend, seasonality or random component(s). They could also look at correlations between major interventions and the fall in mortality rate.
  11. The authors discuss on P3 some of the interventions that have been implemented. Can they narrow down the timeline of implementation of these interventions?

Author Response

Reviewer’s comments (in blue) and Answers (in black)

We want to thank the reviewers for their insightful comments on our paper. We feel the comments have helped us significantly improve the manuscript.

Reviewer: 2

We wish to express our strong appreciation to the reviewers for their insightful comments, which have helped us significantly improve the paper.

Reviewer comments.

  1. P2 L51: Do the authors mean “estimation of the effectiveness of injury prevention measures” instead of “estimation of injury prevention measures”?

Response: We revised this expression as suggested.

  1. P2 L72: The authors indicate that they included only patients with ISS >= 3 in accordance with a previous paper. Can they clarify the rationale for this threshold in a few words in this paragraph?

Response: Thank you for your valuable comment. Although patients with ISS < 3 were excluded from a previous study, we were unable to find out the rational reason for the exclusion of these patients. Therefore, we have redone statistical analysis including paediatric injured patient with ISS < 3.

  1. The authors excluded patients who were dead on arrival, which is somewhat justifiable since they are looking at in-hospital mortality. However, I think it would be useful to show the trend in pre-hospital mortality over time maybe in a supplementary table or figure. It is possible that in-hospital mortality is decreasing at the same time that pre-hospital mortality is increasing? It would be worthwhile proving or disproving this with their

Response: Thank you for your valuable comment. Since it was unclear whether patients with cardiac arrest on hospital arrival died, were declared dead in hospital, or survived, we were unable to calculate the trend in pre-hospital mortality. Since we could not justify exclusion of cases with cardiac arrest on hospital arrival, we have edited the Methods and Results of the additional work suggested by the reviewers. We have redone statistical analysis for paediatric injured patients including cardiac arrest on arrival (N: 392, 2.4% of the total).

  1. Likewise, the authors excluded patients with missing data. What was the pattern of missing data? Did the amount of missing data change over the 10-year period? If it did, e.g. if by the end of the period the TRISS could be calculated on a higher proportion of patients than at the beginning, this may skew mortality estimates (ascertainment bias).

Response: Thank you for your comment. Missing data rates over the 10-year period, are shown on Table S1. We have presented the results in Method section as follows:

Table S1 presents the trend in missing data rates during the study-period. There were no significant statistical differences in the missing data rates by year using a Cochran-Armitage test.

  1. P3 L96-98: I do not understand this sentence. It should be reworded, e.g. “the Cochran-Armitage test was used to test for trends in SMR and in-hospital mortality by study year, which was treated as an ordinal variable.”

Response: We have reworded the sentence as you recommended.

  1. The authors presented the median and IQR for continuous data. My personal preference is to use the median and 5th – 95th percentiles since they give a better indication of the population distribution than the IQR.

Response: Accordingly, we have revised the expression in the main text and Table 1.

  1. P4 L104: What does “mean ... for categorical variables” mean?

Response: Thank you for your comment. We have reworded this expression as follows:

patients’ number and percentages for categorical variables

  1. P4 L115-117: Why were these age thresholds selected? I think the authors should rename “0 years” as “<1 year”.

Response: As suggested, we have revised from 0 years to <1 year in the text, Tables, and Figures.

  1. The authors should present transfusion data, If available, in Table 1 since the requirement for blood products is an independent predictor of trauma mortality. Has blood product utilization changed in this patient population over the ten-year period?

Response: Thank you for your valuable comment. We agree that the requirement blood transfusion might be an independent predictor of trauma mortality. To make this point clearer, we have added the analysis about the trend of urgent blood transfusion within 24 hours after hospital arrival and the regression analysis including urgent blood transfusion as a predictor for mortality risk. The trend in the number of patients needing blood transfusions was not changed, however, patients who needed urgent blood transfusion had a higher mortality risk. We have edited these results in the text and Tables 1 and 2.   

  1. The authors have lost some of the granularity of their data by using only the year of death in the regression model. It would be interesting to see a time series analysis of the data using day or month of death, and then assessing for any trend, seasonality or random component(s). They could also look at correlations between major interventions and the fall in mortality rate.

Response: We agree that the additional information on a time series analysis would be valuable. Regrettably, however, because of the relatively small study cohort by age-group especially of patients <1 year, we did not perform a time series analysis in this study. We are now investigating this point and intend to report on it in future work.   

  1. The authors discuss on P3 some of the interventions that have been implemented. Can they narrow down the timeline of implementation of these interventions?

Response: As suggested, we have added the timeline of implementation of our interventions such as off-the job training and JTDB, in the Discussion section.

Round 2

Reviewer 1 Report

Thank you for your sincere responses to the reviews provided, as reflected in the response document and revised manuscript. I feel the changes made have addressed my leading concerns. Overall the written English is improved, but there are still some English language corrections to be made, and I am grateful for the Editorial team in addressing these prior to publication. 

Thank you again for the opportunity to review this paper.